# Three-dimensional strain dynamics govern the hysteresis in heterogeneous catalysis

Aline R. Passos [1✉], Amélie Rochet [1✉], Luiza M. Manente[1], Ana F. Suzana [1,2], Ross Harder[3], Wonsuk Cha [3] & Florian Meneau [1]

Understanding catalysts strain dynamic behaviours is crucial for the development of cost-effective, efficient, stable and long-lasting catalysts. Here, we reveal in situ three-dimensional strain evolution of single gold nanocrystals during a catalytic CO oxidation reaction under operando conditions with coherent X-ray diffractive imaging. We report direct observation of anisotropic strain dynamics at the nanoscale, where identically crystallographically-oriented facets are qualitatively differently affected by strain leading to preferential active sites formation. Interestingly, the single nanoparticle elastic energy landscape, which we map with attojoule precision, depends on heating versus cooling cycles. The hysteresis observed at the single particle level is following the normal/inverse hysteresis loops of the catalytic performances. This approach opens a powerful avenue for studying, at the single particle level, catalytic nanomaterials and deactivation processes under operando conditions that will enable profound insights into nanoscale catalytic mechanisms.

[1] Brazilian Synchrotron Light Laboratory (LNLS), Brazilian Center for Research in Energy and Materials (CNPEM), 13083-970 Campinas, SP, Brazil. [2] Instituto de Química, UNESP, Rua Professor Francisco Degni, 14800-900 Araraquara, SP, Brazil. [3] Advanced Photon Source, Argonne National Laboratory, 9700 South Cass Avenue, Argonne, IL 60439, USA. ✉email: aline.passos@lnls.br; amelie.rochet@lnls.br

Chemical properties of supported metallic catalysts can be modified by lattice strain that alters the reactivity of metal surfaces[1–4]. As shown theoretically on extended surfaces, the adsorption and dissociation energies can be optimised for enhancing a particular chemical reaction by neatly controlling the degree of lattice surface strain[5]. This is explained by the d-band model[2], with a change in the surface d-band centre due to lattice distortion. For example, gold (Au) is a late transition metal with a d-band more than half-filled. Tensile strain leads to a narrowing of the d-band and consequently an increased population of the d-band. For the model CO oxidation reaction, DFT calculations demonstrated that the altered d-band centre and the tensile strain enhance the adsorption of molecular oxygen on Au surfaces and lowers the dissociation barrier of $CO^{2,6}$.

In nanoparticles (NPs), strain can emanate from intrinsic factors such as nanoparticle's size, morphology, exposed crystallographic facets, crystalline defects and the material itself[3,7–9]. Besides, extrinsic strain can, for example, emerge from lattice mismatch induced at interfaces, or from nanoparticle-support interface or being due to core-shell structures. Strain information are mostly obtained by X-ray diffraction and high-resolution transmission electron microscopy techniques. While X-ray diffraction provides average information[3], atomic resolution is achieved using aberration-corrected transmission electron microscopy (AC-TEM) such as the work of Walsh et al.[10] showing local strain variations due to the nanoparticle-support interface. Under reaction conditions, the nanoparticle structure can present dynamic restructuration, faceting process[11–14], requiring in situ and operando imaging tools[15]. Although the development of environmental TEM enables to image catalysts under reactive conditions, as demonstrated by Vendelbo et al.[11] by in situ TEM following the oscillatory behaviour of Pt nanoparticles during CO oxidation reaction, three-dimensional strain information is lacking. In situ Bragg coherent diffraction imaging (BraggCDI) recently demonstrated the possibility to obtain 3D strain maps and defects dynamic information[16,17]. It has been successfully employed to reveal in situ nanocrystals deformations, bulk and surface strain dynamics, localisation of active sites under operating conditions[12,18–22]. In our previous work[12], we revealed by using in situ BraggCDI, the dynamic faceting of 120 nm gold nanoparticles supported on $TiO_2$ during the catalytic reaction of CO oxidation and the formation of nanotwin defective network to accommodate the strain built up under reaction conditions.

Here we designed a model system where we tune the intrinsic strain, by shaped controlled synthesis of 60 nm gold nanocrystals. Gold cuboctahedra and nanocubes are investigated by operando BraggCDI enabling site-specific strain mapping under CO oxidation reaction cycle. Besides often being considered as a prototypical reaction to study the fundamental concepts of heterogeneous catalysis, this reaction is also of high environmental and societal importance[23]. In particular, it can show inverse and direct hysteresis behaviours during light-off and light-out as reported by Casapu et al.[24]. We seek to explore how the evolution of defect structures and strain in morphology-controlled gold nanoparticles play a role in the catalytic activity during CO oxidation cycle. We observe a correlation between anisotropic strain and the hysteresis behaviour during $CO_2$ production. Our results reveal the formation of anisotropic tensile strain patterns at the surface of the nanocrystal which propagates into the interior during the CO oxidation reaction. The gold nanocube structure enables to clearly evidence the facet-dependent reactivity where identical facets do not have equivalent catalytic response. By mapping the energy landscape with attojoule resolution, we reveal that the hysteresis is at the single-particle level involving the three-dimensional strain field.

## Results

**Morphologies of gold nanocrystals.** Because of the different amount of reductive agent used in our experiment, a colloidal-based seed-mediated growth method based on the work of Sau et al.[25], we can control and tune the morphology of the Au nanocrystals. The gold seeds are formed in the presence of a surfactant leading to spherical nanoparticles with monodisperse size distribution of $6.0 \pm 0.7$ nm. Gold cuboctahedron-shaped nanocrystals and nanocubes were then grown by addition of an appropriate quantity of seed solution to the aqueous growth solutions containing different concentrations of ascorbic acid enabling to control the degree of edges and corners truncation. The gold nanocrystals were then deposited by impregnation on $TiO_2$ support and dried under air. Figure 1a shows scanning electron microscopy (SEM) images of the supported gold nanocrystals. The detailed synthesis procedures are described in 'Methods'. The cuboctahedron and cube nanocrystals display monodisperse size distributions of $68.0 \pm 8.0$ nm and $63.5 \pm 6.5$ nm, respectively, determined by scanning electron microscopy and small angle X-ray scattering measurements (Supplementary Fig. 1). The nanocube crystals are composed of six {100} facets, the edges are truncated to {110} planes, and the corners are formed by small {111} facets. The cuboctahedron crystals are truncated cubes with higher ratio of {111} to {100} facets.

**Local lattice distortion variations in supported Au NPs.** 3D lattice displacement maps were obtained from BraggCDI

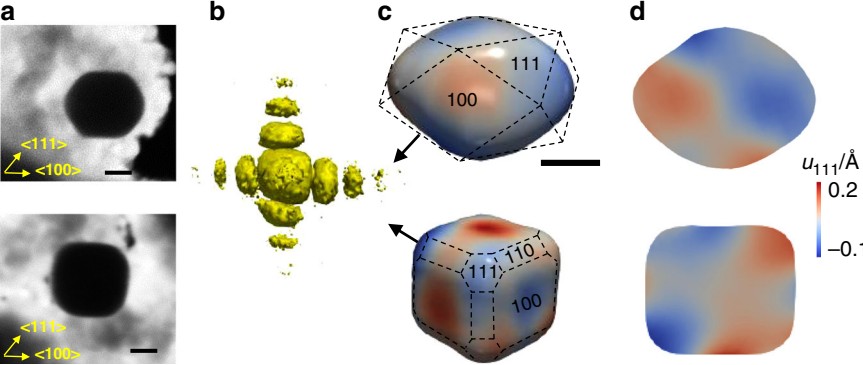

**Fig. 1 Morphology of gold nanocrystals at room temperature. a** Scanning electron microscopy images of the dried supported gold crystals, showing well-defined cuboctahedric (top) and cubic (bottom) shapes. **b** 3D diffraction pattern obtained by rocking scans around the (111) Bragg peak of the cube. **c** 3D displacement field along the [111] direction with the black vector representing the $q_{111}$ scattering vector. The facets orientations are shown. **d** Cross-section views of the internal displacement along the [111] direction. Scale bars, 30 nm.

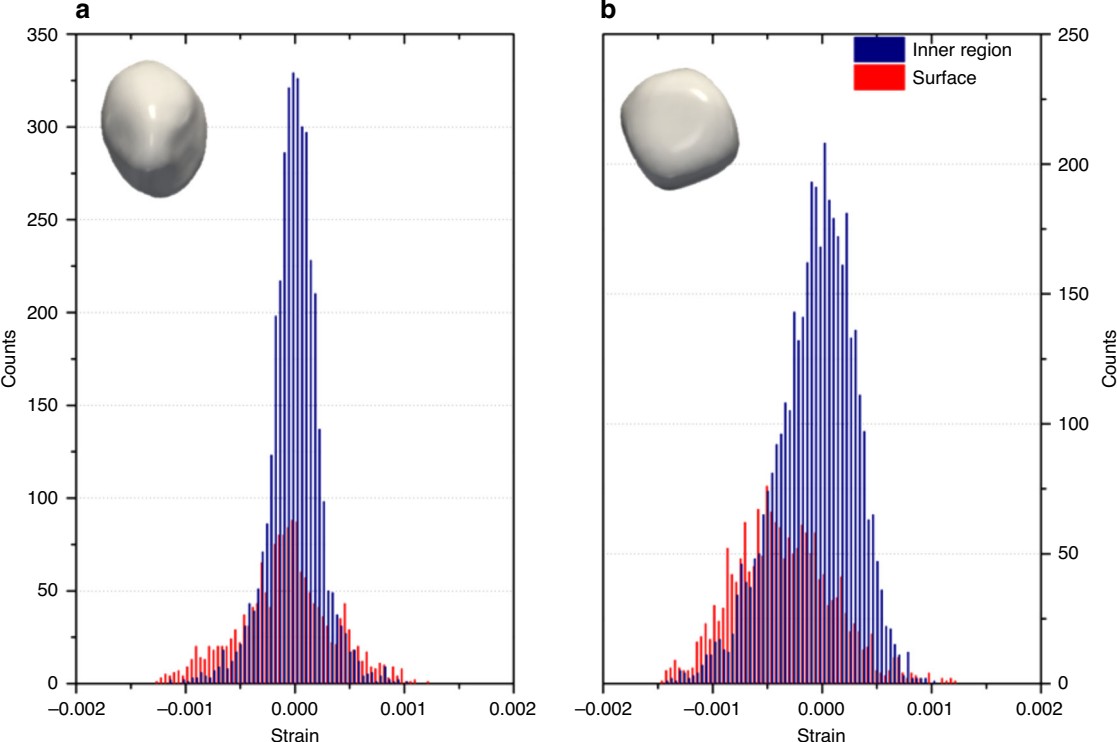

**Fig. 2 Statistical distribution of strain of the calcined nanocrystals. a** Gold cuboctahedron and **b** gold cube. The strain of the inner regions in blue of the nanocrystal is shown with the surfaces strain in red. The corresponding 3D Bragg electron density is shown in insert.

measurements. The pristine supported gold nanocrystals were first imaged in air. In brief, coherent X-ray diffraction patterns were collected around the gold (111) Bragg peak at 9 keV at the 34-ID-C beamline (Advanced Photon Source, USA). The 3D diffraction pattern obtained by rocking scans around the (111) Bragg peak of the cube is presented in Fig. 1b. Using iterative phase retrieval algorithms, the 3D Bragg electron density and the lattice displacement field of the nanocrystals are determined. The detailed coherent X-ray diffraction data analysis is described in the 'Methods'. The amplitude of the reconstructed image represents the electron density and the phase corresponds to the projection of the displacement of the crystal lattice on the scattering vector $q_{111}$. The lattice displacement resolution reaches the picometre level, while the real space resolution 15 nm.

The 3D lattice displacements maps of the cuboctahedron and cube nanocrystals are displayed in Fig. 1c. The nanocrystals were capped with CTAB molecules. The residual distortion in the displacement field observed at the surface for the as-synthetised nanocrystals arises from the growth conditions which are far from equilibrium[26] as shown in Fig. 1c. The size-limiting effect of CTAB affects the growth rate and the lattice deformations across the nanocrystal, from the centre to the outside regions. These inhomogeneities are best visualised in the cross-sections views of the lattice displacement maps shown in Fig. 1d. The images are coloured by the local displacement field where red (positive sign) indicates the projected displacements along the [111] direction, $u_{111}$, and blue (negative sign) implies the opposite direction. The maximum $u_{111}$ value is ~10 and 20 pm of the Au$_{111}$ lattice constant for the cuboctahedron and the cube, respectively. Figure 1d shows that the stress induced by the capped CTAB molecules on gold has a different impact on the flat facets compared to the curved regions of the crystal surface.

Strain along the [111] direction was determined by spatial differentiation of the lattice displacement field[27] $\partial u_{111}/\partial x_{111}$, the

3D strain maps of the nanocrystals are resolved and shown in Supplementary Fig. 2. The 3D strain images enable to visualise and quantify the bulk and surface strains in the NPs, showing strain that can be compressive or tensile. The strain sensitivity is of the order of $\sim 2 \times 10^{-4}$ due to the strong sensitivity of X-rays to the crystal lattice spacing. Such a strain sensitivity is nowadays routinely achieved using BraggCDI methods as reported by[28–31]. It is most pronounced at edges {110} and corner sites {111}, in the case of the gold cube. These 3D strain maps clearly reveal the anisotropic strain distribution of both nanocrystals at room temperature (RT), and a mean surface strain that is compressive for the as-prepared dried cube and tensile for the cuboctahedron. Upon calcination at 400 °C and removal of the CTAB molecules, both nanocrystals display a compressive surface strain, as expected for tensile surface stress of metals[32,33]. Only the map of the [111] component of the lattice strain tensor is presented. Indeed, to determine the full strain tensor, at least three or more non-parallel reflections must be collected. Multiple reflections from a single particle have been measured previously using BraggCDI[29,31,34] but such a study could not be carried out in this work due to the in situ experimental limitations of the operando cell. However, the 111 component is also sensitive to 100, 110 lattice distortions, and so used throughout as a signature of the phenomenon occurring at the surface of the nanocrystal during the catalytic process, to image their elastic response[35]. The projected strain measurements are quantitatively summarised in histograms, in Fig. 2, for the two calcined nanocrystals. The histograms compare the strain distribution from the inner regions (blue) of the NPs to that of their surfaces (red). As anticipated, the cuboctahedron crystal exhibits the lowest strain structure, with a mean compressive surface strain value of $-7.83 \times 10^{-5}$, while the cube crystal mean surface strain value rises up to $-3.52 \times 10^{-4}$. This demonstrate that the strain range of the NPs spans over an order of magnitude depending solely on the nanocrystal morphology.

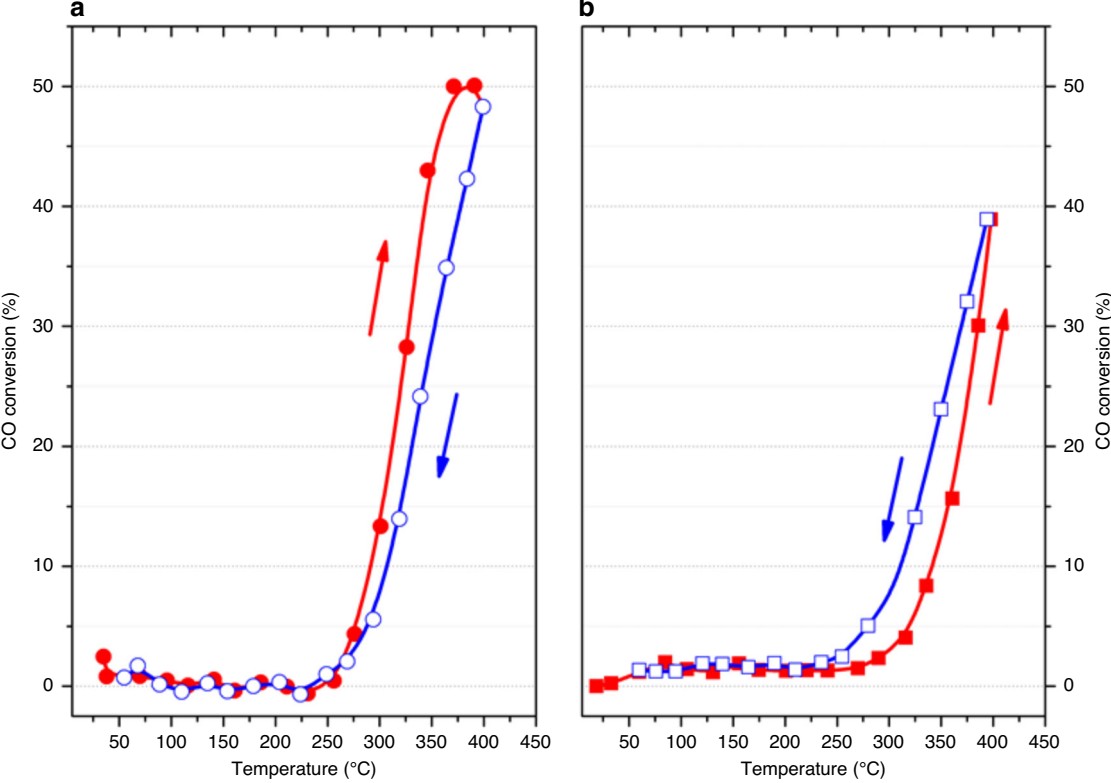

**Fig. 3 Catalytic activity during CO oxidation reaction.** CO conversion as function of temperature for **a** the cuboctahedra and **b** cubes showing the inverse and normal hysteresis loops behaviours, respectively. The red symbols indicate the heating ramps, while the blue symbols indicate the cooling step.

**Effects of strain on reaction kinetics**. To further study the impact of strain in nanoparticles on their catalytic properties, we performed operando BraggCDI measurements during the CO oxidation reaction. The gold nanocrystals in our study are larger than optimum size range where gold is considered to be catalytically active. Nonetheless, they do present catalytic activity at high temperature as shown in Fig. 3 presenting the CO conversion as function of temperature of the cuboctahedra and cubes nanocrystals. We observe typical temperature hysteresis loops, characteristic of the CO oxidation reaction. However, both gold morphologies present distinct hysteresis loops: inverse and normal hysteresis, respectively. For a normal hysteresis, the conversion of CO into $CO_2$ is higher during the cooling step (light-out curve), while for the inverse hysteresis, the conversion of CO is higher during the heating step (light-off curve). The latter results evidence the impact of the nanocrystal morphology and their strain distribution on the hysteresis profile and their catalytic properties.

We further demonstrate that the hysteresis loops correlate with the nanocrystals surface strain dynamics. The detailed experimental conditions regarding gas environment of the operando BraggCDI measurements are described in 'Methods'. Figure 4a, b presents the 3D images and cross-sectional images from the BraggCDI patterns of the gold cuboctahedron (AuNP1). The dashed plane represents the location of the cross-section. We observe a mean surface compressive to tensile strain swap (from $-7.83 \times 10^{-5}$ to $1.40 \times 10^{-5}$) upon flowing the $CO/O_2$ gas mixture on the gold nanocrystal surface, in agreement with Suzana et al.[12].

The changes of the compressive (blue) and tensile (red) strains are shown at the different steps of the reaction. We observe an initial tensile strain located on the {111} facets, which expands as the temperature rises. At 200 °C, the tensile strain propagates

across the nanocrystal, and enlarges through the connection of the opposite {111} facets. This is best visualised in Fig. 4a, which displays the 3D highly compressive and tensile strain evolutions of the nanoparticle during the CO oxidation reaction. We notice the formation of a tensile-strained corona ($>1.0 \times 10^{-4}$, in red) around the nanocrystal counterbalanced by two compressive strained regions ($<-1.3 \times 10^{-4}$, in blue). The tensile strain keeps building up until 400 °C, when the maximum CO conversion is attained. It is well established that the lattice strain can change the reactivity of metal surfaces[1–3]. This is explained by the d-band model, where the tensile strain (regardless of direction in the lattice) leads to a narrowing of the d-band and an increased population of the latter, for late transition metals. The altered d-band centre affects the adsorption and dissociation energies. Accordingly, the tensile strain areas indicate the localisation of the active sites during catalytic CO oxidation reaction.

These tensile-strained areas from the surface extend to inside the nanocrystal during the catalytic process, and revert after the reaction is complete (Fig. 4). During the cooling process, the tensile strain is effectively released and exhibits similar anisotropic strain patterns as during the heating process. Surprisingly, the surface strain patterns are temperature-offset between the heating and the cooling processes coinciding with the hysteresis behaviour of the nanocrystal. As clearly shown in Fig. 4b, the inner strain pattern at 300 °C during cooling matches the one at 200 °C during heating, the 200 °C cooling pattern is equivalent to the 100 °C heating one, and finally the 100 °C and RT cooling present the similar four-fold pattern of the 100 °C heating, as expected for an inverse hysteresis loop. This is quantitatively summarised in the strain histograms in Supplementary Fig. 3, comparing the strain distribution from the inner regions to the surface strain distribution of the cuboctahedron. The strain histograms emphasise the occurrence of highly tensile-strained

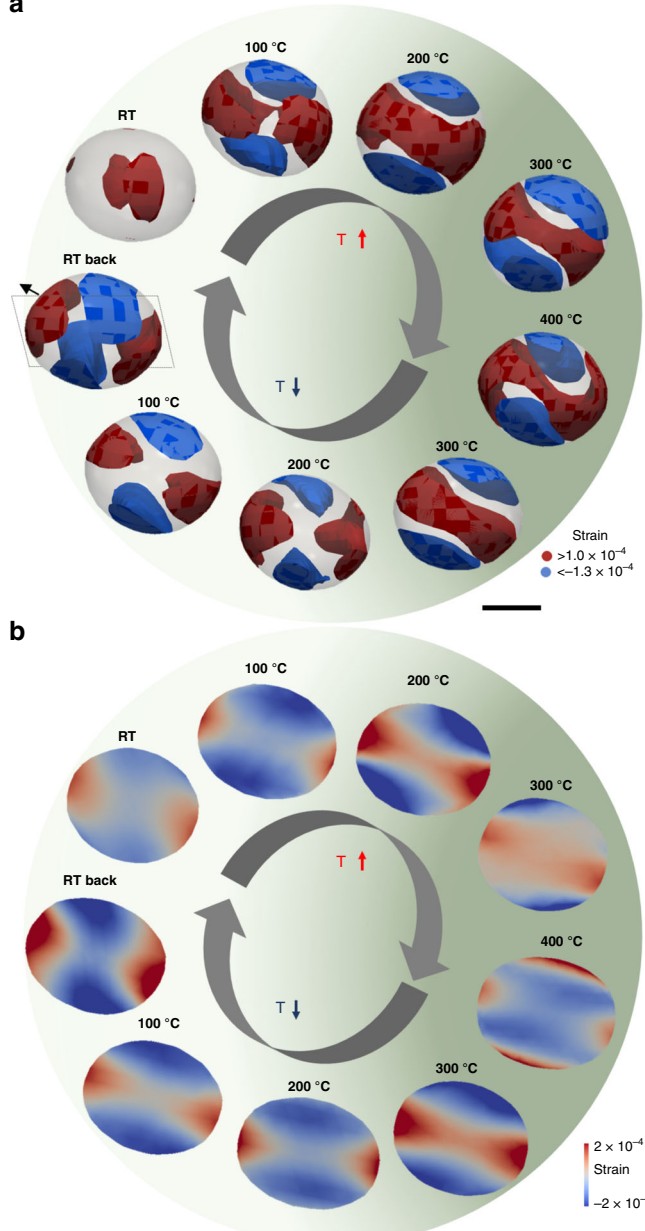

**Fig. 4 Operando 3D strain images of gold cuboctahedron crystal. a** Strain images (strain field projected along (111)) for the highly compressive and tensile strain distribution of the same AuNP1 nanoparticle during CO oxidation reaction at RT, 100, 200, 300 and 400 °C, during heating and cooling steps. Highly compressive (blue, strain < −0.00013) and tensile (red, strain >0.00010) strain regions present anisotropic patterns. The particle shape is shown as a semi-transparent grey isosurface. **b** Corresponding particle cross-sections views of the internal strain field at the dashed line box in (**a**). Scale bar, 30 nm. The green gradient is illustrating the increase/decrease of catalytic activity with the temperature simultaneously followed by mass spectrometry to the BraggCDI experiment.

cube surface strain dynamics present a hysteresis behaviour, coinciding with the $CO_2$ production. The evolution of the surface strain dynamics is displayed in Fig. 5 demonstrating the normal hysteresis behaviour of the cube (Fig. 3b). The largest tensile strain values are primarily concentrated at the {110} edges and {100} facets and less pronounced at the {111} corners. This is in good agreement with DFT calculations for faceted gold nanoparticles showing that the {100} and {110} are the more reactive for oxidation reaction than the {111} which have higher coordination numbers[8].

Outstandingly, the tensile strain at 400 °C, pointing to the active sites, is restricted to four of the six {100} facets of the cube, the remaining two show a compressive strain (Fig. 5c). Although facet-dependent reactivity is well-known, this is the first proof of unequal reactivity of identical facets in a single nanocrystal. This is quantitatively summarised in the strain histograms in Fig. 5d, comparing the surface strain distributions of the six {100} facets.

**Elastic energy landscape**. To elucidate the mechanism driving the hysteresis response of the cuboctahedron and the cube, we determine the elastic energy landscape using the three-dimensional strain distribution to assess the elastic energy which is defined as (Eq. (1)):

$$E_{S} = \frac{3}{2} K \int \left( \frac{\partial u_{111}}{\partial x_{111}} \right)^2 dV \qquad (1)$$

where $K$ is the bulk modulus of Au, $u_{111}$ is the displacement value along [111], and $x_{111}$ is the (111) lattice constant of Au. The strain induced by the deviations of atoms from their equilibrium position are accounted by the elastic strain energy, irrespective of the original cause of displacement. Figure 6 shows the values of the elastic energy, on the order of attojoules, at different reaction conditions.

The 3D mapping of the energy landscape unveils the dynamics of AuNP1 showing a distinct behaviour in strain energy between the heating and cooling steps of the CO oxidation reaction. Although the hysteresis behaviour during oxidation reaction is expected from a macroscopic point of view (catalytic conversion)[36], it turns out to be at the single-particle level. This is surprising and could be explained by taking into account losses in the form of irreversible elastic energy release via lattice deformation and heat dissipation due to the exothermicity of the reaction. This is further supported by the value of the elastic energy of AuNP1 at RT after a CO oxidation cycle which shows that the nanocrystal does recover neither its strain energy (green symbol in Fig. 6) nor its initial strain pattern (Fig. 4). This is clear indication of losses in the form of irreversible elastic energy and explanation to the inverse hysteresis behaviour. On the other hand, the cube nanocrystal shows no variation of strain energy during the catalytic reaction, recovering its initial strain pattern and energy at RT after a cycle. This is compatible with a normal hysteresis behaviour where the pre-strained state of the cube (an order of magnitude higher than that of the cuboctahedron at RT) prevents from deformation and thus energy losses.

In conclusion, using operando BraggCDI, we studied at the single-particle level, the strain dynamics of morphology-controlled gold nanocrystals during the CO oxidation reaction. We revealed their anisotropic strain formation and propagation, and experimentally unveiled that identically crystallographic oriented facets in a single nanocrystal can be differently affected by strain leading to "identical-facet"-dependent reactivity. This anisotropic strain may also provide an explanation for shape modifications during CO oxidation reaction[37]. Finally, we followed the evolution of the elastic energy landscape with attojoule energy resolution and found out that the catalytic

surface regions at 400 °C and further confirm the inverse hysteresis behaviour.

We also imaged a gold cube (AuNP2) under operando CO oxidation reaction (Fig. 5). During the heating process, under reactive conditions, we also observe the formation of highly surface strained regions, which peak at 400 °C, corresponding to the highest CO conversion. Similarly to the cuboctahedron, the

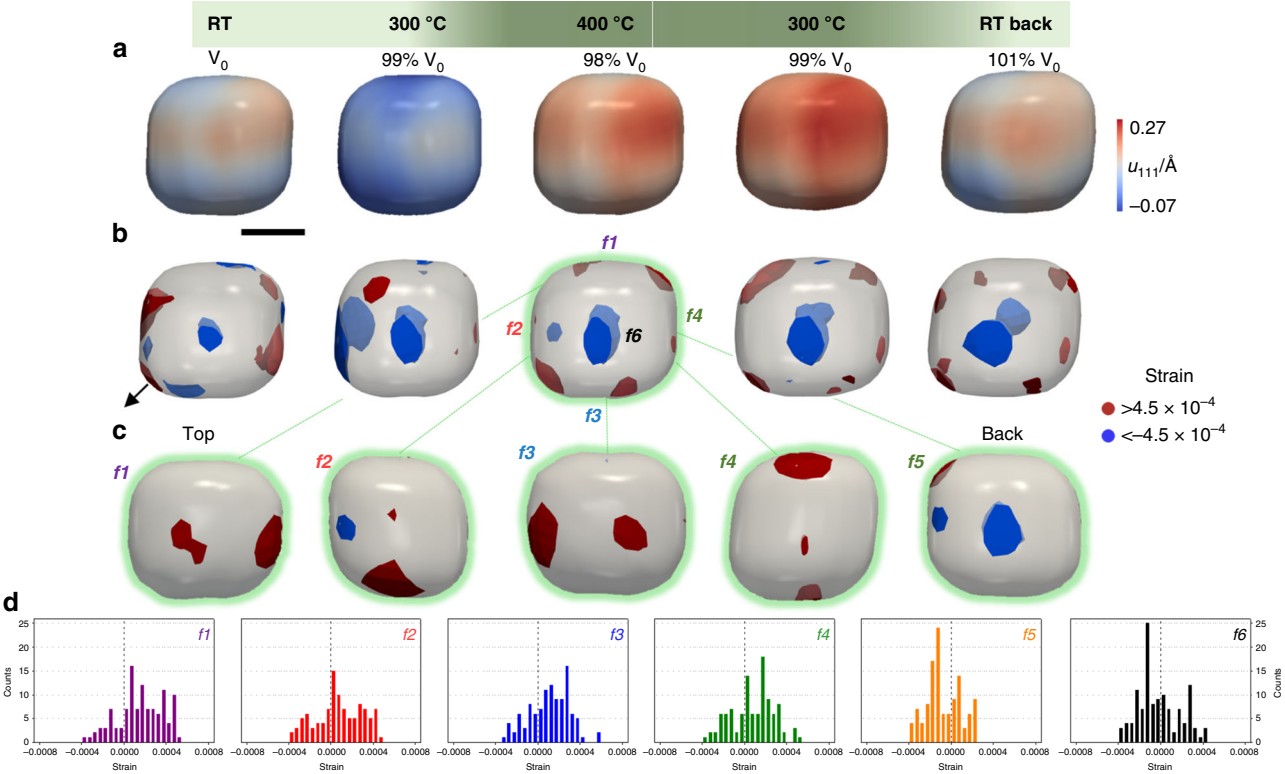

**Fig. 5 Operando 3D distribution of the displacement field during CO oxidation reaction. a** Displacement field along (111) of the same AuNP2 nanoparticle under $CO/O_2$ at RT, 300 and 400 °C, during the entire catalytic hysteresis loop. **b** Corresponding 3D strain structure of highly compressive (blue, strain < −0.00045) and tensile (red, strain > 0.00045) strained regions. The black arrow is indicating the orientation of the $q_{111}$ scattering vector. The particle shape is shown as a semi-transparent grey isosurface. **c** 3D strain distribution at 400 °C with maximum of compressive (blue) and tensile (red) strains regions of the six {100} facets of the cube (f1 to f6). **d** Statistical distribution of strain of the cube facets (f1 violet, f2 red, f3 blue, f4 green, f5 orange and f6 black). Scale bar, 30 nm. The green gradient is illustrating the increase/decrease of catalytic activity.

hysteresis is occurring at the single-particle level. We discovered the origin of the hysteresis and demonstrated that it is linked to the structural properties and strain dynamics of the catalysts and not only to the variations of reaction conditions as long believed. This suggests that finding ways to manipulate the elastic energy is of paramount importance for tuning the chemical and catalytic properties of nanomaterials.

## Methods

**Gold nanocrystals synthesis**. In a typical synthesis, $TiO_2$ was produced via sol–gel hydrolysis precipitation of titanium isopropoxide. 5.75 mL of $Ti(OPri)_4$ (titanium(IV) isopropoxide) was dissolved in a 14.88 mL of isopropanol, 2.80 mL of Milli-Q water was added drop by drop, under magnetic stirring in an ice-bath at ~3 °C. After 30 min, the resultant solution was poured into a glass container that was then sealed and kept at RT during 24 h for gelation. The wet gel was dried at 50 °C for 48 h, and the calcination was carried out in air by heating at 450 °C for 2 h.

Cube and cuboctahedron-shaped AuNP were prepared by a two-step seed-mediated growth method[25]. In a typical seed synthesis, 0.25 mL of $HAuCl_4.3H_2O$ solution (0.01 mol $L^{-1}$) was mixed with 0.75 mL of cetyltrimethylammonium bromide (CTAB, 0.1 mol $L^{-1}$) diluted with 8.5 mL of Milli-Q water. The $Au^{3+}$ was reduced by the addition of 0.6 mL of ice-cold $NaBH_4$ solution (0.01 mol $L^{-1}$). The brownish solution was obtained after vigorously stirring for 2 min allowing the escape of the gas formed during the reaction. In the growth reaction to produce cubic nanoparticles, 0.3 mL $HAuCl_4.3H_2O$ solution (0.01 mol $L^{-1}$) was added to 2.4 mL CTAB solution (0.1 mol $L^{-1}$) followed by the addition of 0.9 mL ascorbic acid (0.1 mol $L^{-1}$), and 0.5 µL Au seed solutions. The cuboctahedric AuNP were produced in a similar way with lower amount of ascorbic acid (0.45 mL).

The AuNP were supported on $TiO_2$, with BET surface area of 57 $m^2 g^{-1}$ and mean pore size 4.5 nm. Typically, the AuNP solution was added to 100 mg of support and acidified by $H_2SO_4$ (pH ~ 1) to obtain 1% w/w of the catalyst. The slurry was stirred for 2 h after which it was centrifuged, washed with water ~10 times or until mother liquor becomes neutral, dried at 110 °C for 4 h. The remaining solid was calcined in two-steps, 200 °C for 1 h to CTAB decomposition then heated up to 400 °C for 1 h. The nanoparticles morphology, size and size distributions were characterised by high-resolution scanning electron microscopy (FEI Inspect F50) operated at 30 kV in transmission mode with a STEM detector in bright field mode. For STEM analysis, the catalyst powder was dispersed ultrasonically in water and then drop-casted onto a carbon-coated copper grid. The particle size distributions were obtained from the measurement of a hundred nanoparticles (Supplementary Fig. 1). The gold nanoparticles suspensions were also characterised by small angle X-ray scattering, at the SAXS1 beamline of the Brazilian Synchrotron Light Laboratory (LNLS). The X-ray beam energy was set to 8 keV, the Pilatus 300k detector was positioned three metres from the sample, enabling to obtain a q-range spanning from 0.004 to 0.14 $Å^{-1}$. The suspensions were loaded in quartz capillaries (Supplementary Fig. 1).

**Catalytic tests**. The activity measurements were carried out in a tubular reactor with 10 mg of catalyst diluted in 90 mg of quartz powder, quartz wool plugs were used to fix the powder. For CO oxidation, the gas composition was $CO:O_2$, 0.4:4.0% balance in He with a total gas flow of 100 mL $min^{-1}$. Heating ramp of 3 °C/min was used to reach isothermal conditions for the data acquisition. The feed and product gas streams were analysed by a gas chromatograph (GC) (Agilent 490 Micro GC) equipped with a thermal conductivity detector (TCD).

**Bragg coherent X-ray diffraction imaging**. BraggCDI experiments were performed at 34-ID-C beamline at the Advances Photon Source in Argonne National Laboratory, USA. The catalyst powder was dispersed ultrasonically in water and then transferred by drop-casting onto a Si wafer and placed in the operando cell. The coherent diffraction patterns, (111) Bragg condition, were collected with a Timepix detector, with $55 \times 55$ µm² pixel sizes, placed 430 mm away from the sample. The operando reactor was scanned with a 9 keV focused coherent X-ray beam ($600 \times 600$ nm²) until an isolated Bragg peak shined on the detector. The 3D diffraction data were acquired as rocking curves with an angular step of 0.02° and 41 frames of 10 s exposure, with 2 or 5 repetitions. The same nanoparticle was measured under reaction conditions during heating and cooling. Ramps of 3 °C/min were used to reach isothermal conditions for the data acquisition at RT, 100, 200, 300 and 400 °C. During the entire thermal treatment, a $CO/O_2$ gas mixture (CO: $O_2$, 0.4: 4%) was used with a total gas flow of 20 mL·$min^{-1}$. The gas effluent was simultaneously analysed by mass spectrometry (Dycor LC100MS, AMETEK).

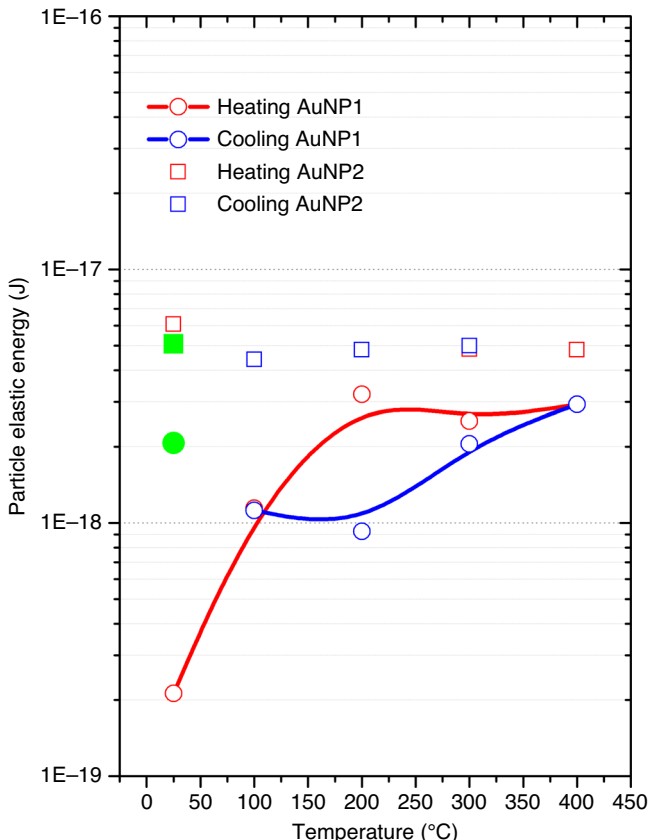

**Fig. 6 Elastic energy landscape under oxidation reaction conditions.**
Elastic energy landscape of a single gold cuboctahedron (AuNP1, solid
lines) and cube (AuNP2, symbols) nanocrystals during heating (red) and
cooling (blue). The green symbols indicate the strain energy at RT after a
CO oxidation cycle. Uncertainties are within the symbols.

**Data reconstruction**. The phase retrieval algorithm was initiated with 20 error-
reduction (ER) iterations[38], followed by 180 iterations of the hybrid input–output
(HIO) algorithm[39], using the guided-approach[40]. 620 iterations were used for the
reconstruction. The support constraint was refined with the shrink-wrap method
and partial coherence is also taken into account[41]. The 3D Bragg electron density as
well as the 3D lattice displacement field projected along $q_{111}$ vector are obtained by
reconstructing the 3D diffraction data. Paraview (http://www.paraview.org) was
used to visualise the reconstructions in two- and three-dimensions.

The strain uncertainty[30,31,42] was determined by only considering material in
the core of the nanocrystals ~40 nm away from the surface. The strain uncertainty
is $1.54 \times 10^{-4}$ and $1.32 \times 10^{-4}$ for the cuboctahedron and the cube nanocrystals,
respectively.

## Data availability
The data supporting the findings of this study are available from the corresponding
author upon request.

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

## Acknowledgements

This research was supported by the São Paulo Research funding agency (FAPESP, grant numbers: 2014/25964-5, 2014/27127-3, 2017/23050-4, 2018/08816-3, 2019/03162-8). The Advanced Photon Source (APS) is acknowledged for providing beamtime at the 34-ID-C beamline (proposal GUP-56879). The LNLS is acknowledged for access to the SAXS1 beamline and the High-Performance Computing system for performing the reconstruction of the BraggCDI data (proposals 20180704, 20190268) and the LNNano for technical support during electron microscopy work (proposal SEM-24340, SEM-C1-25076). This research used resources of the Advanced Photon Source, a U.S. Department of Energy (DOE) Office of Science User Facility operated for the DOE Office of Science by Argonne National Laboratory under Contract No. DE-AC02-06CH11357.

## Author contributions

A.R.P., A.R., A.F.S., R.H. and W.C. carried out the BraggCDI imaging experiments at 34-ID-C beamline. A.R.P., A.R. and F.M. performed the CXD data analysis. A.R.P prepared the gold nanocrystals and performed with A.R. and L.M.M. the materials characterisation and catalytic tests. A.R.P., A.R. and F.M. conceived the project and wrote the manuscript with input from all authors.

## Competing interests

The authors declare no competing interests.
