## [Peer Review File · Nature Communications]

REVIEWER COMMENTS

Reviewer #1 (Remarks to the Author):

The 3D strain distribution in catalytic nanoparticles is important for the understanding the catalytic activity. The variation in 3D strain distribution as a function of temperature combined with the hysteresis in catalytic activity upon heating and cooling adds even further understanding of the catalytic activity under operation conditions. This manuscript provides important insights that are worthy of publication in Nature Communications. However, the manuscript needs further and necessary clarifications prior to publication. The following initial questions and comments should be addressed.

Page 2, second paragraph, last sentence. Please provide a complete sentence.

Page 4. The authors claim that the two types of nanoparticles had cuboctahedron and cubic morphologies. Figure 1 shows the cubic morphology but the cuboctahedron morphology is not clearly seen in Figure 1c. Why?

Page 4. The authors claim that the two types of gold nanoparticles had monodispersed size distributions of 68.0 ± 8.0 nm and 63.5 ± 6.5 nm. Please provide information about how the size distribution was determined.

Page 5, Figure 1a. Which signal has been used to obtain the scanning electron microscopy images? Why is the particle dark compared to the background?

Page 6, first line. The lattice displacement resolution is said to reach tens of picometers and the real space resolution is 15 nm. Does this mean that the x-ray data shown in Figures 3, 4, 5 and 6 has a spatial resolution of 15 nm over the particles that are about 60-70 nm in diameter?

Page 6, second and third paragraph. The text says that the maximum u_{111} value is about -100 to 200 pm. The Au_{111} lattice constant at room temperature is about 236 pm. A displacement of 200 pm should include both the effect of the thermal expansion and the displacement due to strain. The resolution of the displacement measurements should be tens of picometers, as stated earlier on page 6, and this corresponds to about 0.08 strain. The authors need to clarify how the strain maps were derived using the derivative of the displacement field and provide an estimate of the accuracy of the strain values.

Page 10, first paragraph. It is clear from the subsequent text that NP1 is assumed to have a cuboctahedron morphology. This could be clarified when NP1 is mentioned for the first time in the manuscript. The cuboctahedron morphology is not clear in Figures 4 and 5. Please explain why.

Page 11, Figure 4. The scale bar is missing.

Page 11, Figure 4 a. There is an asymmetry in the strain between the top and bottom of the nanoparticle

at 400 °C (heating up) and at 300 °C, 200°C and 100°C during cooling. Why?

Page 11, Figure 4b. Looking at the strain at the top and bottom surface at 400°C, there is compressive strain in a) but tensile in b). Why?

Page 12, Figure 5. The information provided in Figure 5 is already presented in Figure. I do not see the need for Figure 5.

Page 13, last paragraph. Is there an explanation for the observation that four {100} facets show tensile strain and two compressive?

Page 16, Figure 7. The pink colour is not easily distinguished from the red one.

Page 18, last paragraph. Please explain how an individual Au nanoparticle with a 60-70 nm diameter identified using a 600 x 600 nm² X-ray beam?

Reviewer #2 (Remarks to the Author):

In this work, the authors present in situ three-dimensional strain evolution of single gold nanocrystals during the catalytic CO oxidation reaction under operando conditions with coherent X-ray diffractive imaging. The authors claim a direct observation of anisotropic strain dynamics at the nanoscale where identically crystallographic oriented facets are qualitatively differently affected by strain leading to preferential active sites formation. Specially, the authors found that the single nanoparticle elastic energy depends on heating versus cooling cycles, and the hysteresis observed at the single particle level is following the normal/inverse hysteresis loops of the catalytic performances.

Overall, I think this work is impressive, especially for that the observed strain dynamic hysteresis follow the normal/inverse hysteresis loops of the catalytic performances. I believe the manuscript will be of general interest to the communities of nanomaterials and materials characterization. I would like to recommend the publication of this work after the authors address the following questions / comments.

Questions and comments:

1. The authors claim an anisotropic strain dynamics at the nanoscale where identically crystallographic oriented facets are qualitatively differently affected by strain, as evidenced in Figure 5 and 6. It must be pointed out that the strain map obtained in this work was only the map of [111] component of the lattice strain tensor. Based on these, it is insufficient to evaluate the difference of strain state in different {001} or {111} planes. If the authors want to obtain the complete information of the strain tensor, they should measure more than 3 peaks in the reciprocal space using Bragg CDI. Similarly, the sign of “tensile” or “compressive” strain presented in figure 5 and 6 does not means a big difference in strain state for identically crystallographic oriented facets. The sign of the [111] component of strain is

largely determined by the angle between the crystal plane and the [111] crystalline direction. This point should be clearly stated in the manuscript.

2. The SEM image shown in Fig1a is a bit fuzzy. Can the author present clear pictures to show the morphology and orientation of the NPs? Which can help readers better understand the CDI results through comparison.

3. In Figure 1d, the maximum displacement was shown to be about 0.1-0.2Å. Is this right or wrong? Since the d111 lattice constant of Au is 2.355Å, such big displacement means a strain value is at the level of ~10%. But, in the supplemental Figure 1, the calculated strain values was reported to be 0.15%. These results do not seem to match. Also does not match the maximum displacement of 100-200pm described in Line2, Page 6.

4. The surface displacement of NP affected by CTAB is as high as 200pm. It is difficult to consider it as a strained phase of the internal crystal phase.

5. Strain information along three axis of unit cell is essential for figure out an atomic model for theoretical calculation of electron energy information which is closely related to chemical properties.

6. Although it is close to the best current level of synchrotron imaging, the 15nm spatial resolution seems insufficient to analyze 60nm NPs. Therefore, the 3D strain information obtained for the NPs is relatively rough.

7. Please add scale bar in Figure 4 and 5. Add a caption for Figure 6(d).

Reviewer #1 (Remarks to the Author):

The 3D strain distribution in catalytic nanoparticles is important for the understanding the catalytic activity. The variation in 3D strain distribution as a function of temperature combined with the hysteresis in catalytic activity upon heating and cooling adds even further understanding of the catalytic activity under operation conditions. This manuscript provides important insights that are worthy of publication in Nature Communications. However, the manuscript needs further and necessary clarifications prior to publication. The following initial questions and comments should be addressed.

We thank Reviewer #1 and appreciate his positive feedback.

Reviewer #1

Page 2, second paragraph, last sentence. Please provide a complete sentence.

Author reply:

We modified the sentence: “Besides extrinsic strain can emerge from lattice mismatch induced at interfaces, nanoparticle support interface, core-shell structures...”

by:

Changes in the manuscript:

Besides, extrinsic strain can, for example, emerge from lattice mismatch induced at interfaces, or from nanoparticle-support interface or being due to core-shell structures.

Reviewer #1

Page 4. The authors claim that the two types of nanoparticles had cuboctahedron and cubic morphologies. Figure 1 shows the cubic morphology but the cuboctahedron morphology is not clearly seen in Figure 1c. Why?

Author reply:

The 15 nm spatial resolution of the reconstruction leads to the rounded cuboctahedron shape. However, as stated by reviewer 2, the spatial resolution achieved is “the best current level of synchrotron imaging”. Although this spatial resolution clearly enables to retrieve the shape and size of the gold nanocrystals, in full agreement with the electron microscopy, the edges and corners of the reconstructed objects are not as sharp as the ones observed with electron microscopy. Nonetheless, let’s recall the argumentation developed in the paper is focused on the strain dynamics.

Reviewer #1

Page 4. The authors claim that the two types of gold nanoparticles had monodispersed size distributions of 68.0 ± 8.0 nm and 63.5 ± 6.5 nm. Please provide information about how the size distribution was determined.

Author reply:

The particle sizes distribution was determined by the measurements of 100 particles from the SEM images. Moreover, small Angle X-ray scattering patterns of the nanoparticles in the suspensions were collected showing the high degree of monodispersity. This information is now included in the method section and in a new figure, Supplementary Fig. 1, displaying the histograms obtained from the SEM and the two SAXS patterns.

Changes in the manuscript:

At the end of the sentence: "The cubooctahedron and cube nanocrystals display monodisperse size distributions of 68.0 ± 8.0 nm and 63.5 ± 6.5 nm respectively",

we added: determined by scanning electron microscopy and small angle X-ray scattering measurements (Supplementary Fig. 1).

Changes in the supplementary section:

We added the Supplementary Fig. 1 displaying the histograms obtained from the SEM and the two SAXS patterns with the following figure caption:

Supplementary Fig. 1. Particle size distribution obtained from STEM images of (a) cubooctahedric and (b) cubic shape nanoparticles. (c) Small Angle X-ray Scattering patterns of the gold suspensions, AuNP1 correspond to the cubooctahedra and AuNP2 to the cubes.

Changes in the Methods section:

The nanoparticles morphology, size and size distributions were characterised by high resolution scanning electron microscopy (FEI Inspect F50) operated at 30 kV in transmission mode with a STEM detector in bright field mode. For STEM analysis, the catalyst powder was dispersed ultrasonically in water and then drop-casted onto a carbon-coated copper grid. The particle size distributions were obtained from the measurement of a hundred nanoparticles (Supplementary Fig. 1). The gold nanoparticles suspensions were also characterised by Small Angle X-ray Scattering, at the SAXS1 beamline of the Brazilian Synchrotron Light Laboratory (LNLS). The X-ray beam energy was set to 8 keV, the Pilatus 300k detector was positioned three meters from the sample, enabling to obtain a q-range spanning from 0.004 to 0.14 Å⁻¹. The suspensions were loaded in quartz capillaries (Supplementary Fig. 1).

Reviewer #1

Page 5, Figure 1a. Which signal has been used to obtain the scanning electron microscopy images? Why is the particle dark compared to the background?

Author reply:

The nanoparticles were characterised by high resolution scanning transmission electron microscopy (STEM) in bright field mode to enhance the contrast between the gold nanoparticle and the oxide support. The regions with heavier atoms (gold nanoparticles) are darker, while the support (titanium dioxide) are brighter.

Changes in the methods section:

The nanoparticles morphology, size and size distributions were characterised by high resolution scanning electron microscopy (FEI Inspect F50) operated at 30 kV in transmission mode with a STEM detector in bright field mode. For STEM analysis, the catalyst powder was dispersed ultrasonically in water and then drop-casted onto a carbon-coated copper grid. The particle size distributions were obtained from the measurement of a hundred nanoparticles (Supplementary Fig. 1).

Reviewer #1

Page 6, first line. The lattice displacement resolution is said to reach tens of picometers and the real space resolution is 15 nm. Does this mean that the x-ray data shown in Figures 3, 4, 5 and 6 has a spatial resolution of 15 nm over the particles that are about 60-70 nm in diameter?

Author reply:

This is correct, the figures presented herein have spatial resolution of 15 nm. As stated by reviewer 2, the spatial resolution achieved is “the best current level of synchrotron imaging”. Still, this spatial resolution clearly enables to retrieve the shape and size of the gold nanocrystals, in full agreement with the electron microscopy.

Reviewer #1

Page 6, second and third paragraph. The text says that the maximum u_{111} value is about -100 to 200 pm. The Au₁₁₁ lattice constant at room temperature is about 236 pm. A displacement of 200 pm should include both the effect of the thermal expansion and the displacement due to strain. The resolution of the displacement measurements should be tens of picometers, as stated earlier on page 6, and this corresponds to about 0.08 strain. The authors need to clarify how the strain maps were derived using the derivative of the displacement field and provide an estimate of the accuracy of the strain values.

Author reply:

We thank both reviewers who spotted these errors. These are indeed typographic errors and it should read 10 and 20 pm instead of 100 and 200 pm. The displacements of 10 and 20 pm reported here, do not include the effect of thermal expansion, as measured at room temperature, but relate solely to the stress induced by the capped CTAB molecules. The strain was determined by spatial differentiation of the lattice displacement field $\partial u_{111}/\partial x_{111}$, Hruszkewycz *et al.* (2017) while the strain accuracy is controlled by the strong sensitivity of X-rays to the crystal lattice spacing. Strain sensitivity of 10^{-4} is nowadays routinely achieved using BraggCDI methods as reported by:

- S. O. Hruszkewycz *et al.* *In situ study of annealing-induced strain relaxation in diamond nanoparticles using Bragg coherent diffraction*, *APL Materials*, 5 026105 (2017).
- F. Hofmann, *et al.*, *3D lattice distortions and defect structures in ion-implanted nano-crystals*, *Scientific Reports* 7, 45993 (2017).
- N. W. Phillips *et al.* *Nanoscale Lattice Strains in Self-ion-implanted Tungsten*. *Acta Materialia*, 195, 219-228 (2020).

We determined the strain uncertainty by only considering material 40 nm away from the surface, in the core of the nanocrystals (Supplementary Fig. 2). The strain uncertainty is $1.54 \cdot 10^{-4}$ and $1.32 \cdot 10^{-4}$ for the cuboctahedron and the cube nanocrystals, respectively.

Changes in the manuscript:

Page 6, we modified the sentence “*The lattice displacement resolution reaches tens of picometers, while the real space resolution 15 nm*”, by *The lattice displacement resolution reaches the picometer level, while the real space resolution 15 nm*.

We corrected “100 and 200 pm” by *10 and 20 pm*. We added the sentence: *Strain along the [111] direction was determined by spatial differentiation of the lattice displacement field $\partial u_{111}/\partial x_{111}$* . We added the following reference:

- *S. O. Hruszkewycz et al. In situ study of annealing-induced strain relaxation in diamond nanoparticles using Bragg coherent diffraction, APL Materials, 5 026105 (2017).*

We modified Supplementary Fig. 1 by Supplementary Fig. 2.

We also added the sentence: *The spatial resolution of the reconstructions reaches 15 nm, while the strain sensitivity is of the order of $\sim 2 \cdot 10^{-4}$ due to the strong sensitivity of X-rays to the crystal lattice spacing. Such a strain sensitivity is nowadays routinely achieved using BraggCDI methods as reported by:*

- *A. Ulvestad et al. Three-dimensional imaging of dislocation dynamics during the hydriding phase transformation. Nat. Mater. 16, 565–571 (2017).*

- *F. Hofmann et al. 3D lattice distortions and defect structures in ion-implanted nano-crystals, Scientific Reports 7,45993 (2017).*

- *N.W. Phillips et al. Nanoscale Lattice Strains in Self-ion-implanted Tungsten. Acta Materialia, 195, 219-228 (2020).*

- *F. Hofmann et al. Nano-scale imaging of the full strain tensor of specific dislocations extracted from a bulk sample, Phys Rev Mat, 4, 013801, (2020).*

We added in the Methods section of Bragg coherent X-ray diffraction imaging, the following sentence: *The strain uncertainty was determined by only considering material in the core of the nanocrystals ~ 40 nm away from the surface. The strain uncertainty is $1.54 \cdot 10^{-4}$ and $1.32 \cdot 10^{-4}$ for the cuboctahedron and the cube nanocrystals, respectively.*

We added the references:

- *F. Hofmann et al. 3D lattice distortions and defect structures in ion-implanted nano-crystals, Scientific Reports 7,45993 (2017).*

- *N.W. Phillips et al. Nanoscale Lattice Strains in Self-ion-implanted Tungsten. Acta Materialia, 195, 219-228 (2020).*

- J. Carnis et al. Towards a quantitative determination of strain in Bragg Coherent X-ray Diffraction Imaging: artefacts and sign convention in reconstructions *Scientific Reports* 9 1-13 (2019).

Reviewer #1

Page 10, first paragraph. It is clear from the subsequent text that NP1 is assumed to have a cuboctahedron morphology. This could be clarified when NP1 is mentioned for the first time in the manuscript. The cuboctahedron morphology is not clear in Figures 4 and 5. Please explain why.

Author reply:

Yes, AuNP1 has a cuboctahedron morphology. It is true that the cuboctahedron morphology is not very well-resolved in the Figures 4 and 5, and appear more rounded. This is due to the experimental setup, to perform *in situ* BraggCDI measurements, one must make a compromise between the exposure time and time-resolution dictated by the experiment itself. Moreover, due to drifts (thermal/mechanical) during the experiments, the longer the acquisition time, the higher the probability of “loosing” the nanocrystal under study. As a consequence, the resolution achieved becomes limited by the “fast measurements” acquisition mode. As indicated in the methods, we used 10s exposure times with 2 or 5 repetitions. 2 repetitions were used during the heating and cooling measurements while the RT measurements could be performed with 5 repetitions.

Changes in the manuscript:

Page 10, we modified the sentence: “Figure 4 presents the 3D images and cross-sectional images from the BraggCDI patterns of AuNP1” by:

Figure 4a and b present the 3D images and cross-sectional images from the BraggCDI patterns of the gold cuboctahedron (AuNP1).

The caption of figure 4 is also modified, adding the word cuboctahedron for clarity.

Reviewer #1

Page 11, Figure 4. The scale bar is missing.

Author reply:

The scale bar is now included.

Reviewer #1

Page 11, Figure 4 a. There is an asymmetry in the strain between the top and bottom of the nanoparticle at 400 °C (heating up) and at 300 °C, 200°C and 100°C during cooling. Why?

Author reply:

This is correct. But this is due to the orientation of the 3D views that indeed do give the impression of an asymmetry in the strain, which is not real. Indeed, having a closer look at the 2D slices, this asymmetry is not present, nor on the 3D views presented in Figure 5. Anticipating the answer to the comment of Reviewer 1, on page 12: indeed the data presented in Figure 5 and Figure 4a are the same, but their representation in 3D are different. We believe that Figure 5 presents best the 3D data, and is best supporting the manuscript, so we combined Figure 4 and 5 to a new Figure 4 a and b as shown below:

We modified the figure caption:

Figure 4 (a) *Operando* 3D strain images (strain field projected along (111)) for the highly compressive and tensile strain distribution of the same AuNP1 nanoparticle (cuboctahedron) during CO oxidation reaction at RT, 100 °C, 200 °C, 300 °C and 400 °C, during heating and cooling steps. Highly compressive (blue, strain < -0.00013) and tensile (red, strain > 0.00010) strains regions present anisotropic patterns. The particle shape is shown as a semi-transparent gray isosurface. (b) Corresponding particle cross-sections views of the internal strain field at the dashed line box in (a). Scale bar, 30 nm. The green gradient is illustrating the increase/decrease of catalytic activity with the temperature simultaneously followed by mass spectrometry to the BraggCDI experiment.

Reviewer #1

Page 11, Figure 4b. Looking at the strain at the top and bottom surface at 400°C, there is compressive strain in a) but tensile in b). Why?

Author reply:

This is correct. This is due to the orientation of the views. Indeed, the slice was taken from the diagonal plane (indicated by the dashed line box), so the top and bottom in the 2D slice do not correspond to the top and bottom in the 3D representation.

Reviewer #1

Page 12, Figure 5. The information provided in Figure 5 is already presented in Figure. I do not see the need for Figure 5.

Author reply:

As previously explained, we combined Figures 4 and 5 to best present the 3D data into a single Figure 4 a and b, see above.

Reviewer #1

Page 13, last paragraph. Is there an explanation for the observation that four {100} facets show tensile strain and two compressive?

Author reply:

One explanation could be that to sustain stability of the nanocrystal, the facets accommodate opposite strain avoiding too large distortions. If all facets were presenting tensile strain, the material would present an auxetic behaviour, which is very unlikely. On the other hand, this anisotropic strain behaviour is most probably a first explanation for the shape changes observed during oxidation processes. We already proposed this explanation in the conclusion.

Reviewer #1

Page 16, Figure 7. The pink colour is not easily distinguished from the red one.

Author reply:

We changed the colour to green.

Reviewer #1

Page 18, last paragraph. Please explain how an individual Au nanoparticle with a 60-70 nm diameter identified using a 600 x 600 nm² X-ray beam?

Author reply:

The gold nanocrystals were randomly distributed on the TiO₂ surface. The catalyst powder was dispersed ultrasonically in water and then transferred by drop-casting onto a Si wafer. The latter was placed in the *operando* cell and scanned until a Bragg peak (111) from a random crystal lights up on the detector. Other nanoparticles might be illuminated by the X-ray beam however only a single one is shining on the detector.

Changes in the manuscript:

This information is now included in the method section: *The catalyst powder was dispersed ultrasonically in water and then transferred by drop-casting onto a Si wafer and placed in the operando cell.*

The operando reactor was scanned with a 9 keV focused coherent X-ray beam (600 x 600 nm²) until an isolated Bragg peak shined on the detector.

Reviewer #2 (Remarks to the Author):

In this work, the authors present in situ three-dimensional strain evolution of single gold nanocrystals during the catalytic CO oxidation reaction under operando conditions with coherent X-ray diffractive imaging. The authors claim a direct observation of anisotropic strain dynamics at the nanoscale where identically crystallographic oriented facets are qualitatively differently affected by strain leading to preferential active sites formation. Specially, the authors found that the single nanoparticle elastic energy depends on heating versus cooling cycles, and the hysteresis observed at the single particle level is following the normal/inverse hysteresis loops of the catalytic performances.

Overall, I think this work is impressive, especially for that the observed strain dynamic hysteresis follow the normal/inverse hysteresis loops of the catalytic performances. I believe the manuscript will be of general interest to the communities of nanomaterials and materials characterization. I would like to recommend the publication of this work after the authors address the following questions / comments.

We thank Reviewer #2 and appreciate his positive feedback.

Reviewer #2

Questions and comments:

1. The authors claim an anisotropic strain dynamics at the nanoscale where identically crystallographic oriented facets are qualitatively differently affected by strain, as evidenced in Figure 5 and 6. It must be pointed out that the strain map obtained in this work was only the map of [111] component of the lattice strain tensor. Based on these, it is insufficient to evaluate the difference of strain state in different {001} or {111} planes. If the authors want to obtain the complete information of the strain tensor, they should measure more than 3 peaks in the reciprocal space using Bragg CDI. Similarly, the sign of “tensile’ or “compressive” strain presented in figure 5 and 6 does not means a big difference in strain state for identically crystallographic oriented facets. The sign of the [111] component of strain is largely determined by the angle between the crystal plane and the [111] crystalline direction. This point should be clearly stated in the manuscript.

Author reply:

Yes, the strain map presented in this work is the map of the [111] component of the lattice strain tensor. We agree with Reviewer 2, this is not sufficient to evaluate the difference of strain state in different {001} or {111} planes and to determine the full lattice strain tensor, where at least three or more non-parallel reflections must be collected.

However, our experiments provide a precise *in situ* measurement of the strain along the [111] direction during the oxidation reaction. With the assumption of a cubic symmetry, it is indeed sufficient to measure one Bragg reflection as reported by Ulvestad *et al.* (Andrew Ulvestad *et al.*, In Situ 3D Imaging of Catalysis Induced Strain in Gold Nanoparticles, J Phys Chem Lett, 7, 3008-3013 (2016)). Multiple reflections from a single particle have been measured previously using BraggCDI but such a study could not be carried out in this work due to the *in situ* experimental limitations of the *operando* conditions.

Changes in the manuscript:

We added the following sentence page 7: *Only the map of the [111] component of the lattice strain tensor is presented. Indeed, to determine the full strain tensor, at least three or more non-parallel reflections must be collected. Multiple reflections from a single particle have been measured previously using BraggCDI but such a study could not be carried out in this work due to the in situ experimental limitations of the operando cell. However, with the assumption of a cubic symmetry, it is sufficient to measure one Bragg reflection.*

We included the following references:

- Newton, M. C. *et al.* Three-dimensional imaging of strain in a single ZnO nanorod. Nat. Mater. 9, 120–124 (2009).
- Hofmann, F *et al.* 3D lattice distortions and defect structures in ion-implanted nano-crystals, Scientific Reports, 7, 45993 (2017)
- Hofmann, F. *et al.* Nanoscale Lattice Strains in Self-ion-implanted Tungsten. Acta Materialia, 195, 219-228 (2020).
- Ulvestad, A. *et al.*, In Situ 3D Imaging of Catalysis Induced Strain in Gold Nanoparticles, J Phys Chem Lett, 7, 3008-3013 (2016)

We also included in the supplementary Fig. 2 the last remark of Reviewer 2: *Note that the sign of the [111] component of strain is largely determined by the angle between the crystal plane and the [111] crystalline direction.*

Reviewer #2

2. The SEM image shown in Fig1a is a bit fuzzy. Can the author present clear pictures to show the morphology and orientation of the NPs? Which can help readers better understand the CDI results through comparison.

Author reply:

We rotated the cuboctahedron image and added the crystallographic directions for clarity. The nanoparticles morphology was characterised by high resolution scanning electron microscopy in transmission mode (STEM), which allows only 2D view.

Reviewer #2

3. In Figure 1d, the maximum displacement was shown to be about 0.1-0.2Å. Is this right or wrong? Since the d111 lattice constant of Au is 2.355Å, such big displacement means a strain value is at the level of ~10%. But, in the supplemental Figure 1, the calculated strain values was reported to be 0.15%. These results do not seem to match. Also does not match the maximum displacement of 100-200pm described in Line2, Page 6.

Author reply:

Yes indeed, as pointed out by Reviewer 1, these are typographic errors and it should read 10 and 20 pm instead of 100 and 200 pm. The scale bars of Figure 1d are correct. The broad value range was chosen for best visualisation of the data, and does correspond with the values reported in the manuscript.

Reviewer #2

4. The surface displacement of NP affected by CTAB is as high as 200pm. It is difficult to consider it as a strained phase of the internal crystal phase.

Author reply:

As previously answered, the true value is 20 pm.

Reviewer #2

5. Strain information along three axis of unit cell is essential for figure out an atomic model for theoretical calculation of electron energy information which is closely related to chemical properties.

Author reply:

As previously indicated, we do agree with Reviewer 2. We would need the strain information along three axis of unit cell in order to build an atomic model for theoretical calculations of electron energy. However, due to experimental limitations we measured a single reflection, which still enables a precise determination of the strain along the [111] direction during the oxidation reaction. Note that this is the first *operando* BraggCDI work, following single nanocrystal through the whole catalytic cycle, of heating and cooling processes. But we fully agree with Reviewer 2 that collecting at least three reflections would benefit to the understanding. With the advent of 4th generation synchrotron and upgraded synchrotrons, these measurements will soon become feasible.

Reviewer #2

6. Although it is close to the best current level of synchrotron imaging, the 15nm spatial resolution seems insufficient to analyze 60nm NPs. Therefore, the 3D strain information obtained for the NPs is relatively rough.

Author reply:

We indeed achieved 15 nm spatial resolution, and although this spatial resolution clearly enables to retrieve the shapes and sizes of the gold nanocrystals, in full agreement with the electron microscopy, the edges and corners of the reconstructed objects are not as sharp as the ones observed with electron microscopy. Nonetheless, the argumentation developed in the paper is focused on the strain dynamics, which are revealed with high accuracy.

Reviewer #2

7. Please add scale bar in Figure 4 and 5. Add a caption for Figure 6(d).

Author reply:

We included the scale bar and a caption for Figure 5d (previously Figure 6d)

Changes in the manuscript:

We added to Figure 5 (previously Figure 6) caption: (d) Statistical distribution of strain of the cube facets.

REVIEWERS' COMMENTS:

Reviewer #2 (Remarks to the Author):

The authors have addressed almost all the points that I have raised. I have another two suggestions.

For the d-band theory, I think, the author should explain that whether this theory is related to the average lattice strain in three directions of the surface part of the grain? Or is it related to the average lattice strain along in-plane directions of the surface part of the grain?

“Only the map of the [111] component of the lattice strain tensor is presented. Indeed, to determine the full strain tensor, at least three or more non-parallel reflections must be collected. Multiple reflections from a single particle have been measured previously using BraggCDI, but such a study could not be carried out in this work due to the in situ experimental limitations of the operando cell. However, with the assumption of a cubic symmetry, it is sufficient to measure one Bragg reflection”

I agree to the viewpoint that “with the assumption of a cubic symmetry, it is sufficient to measure one Bragg reflection”. According to this viewpoint and the answer to the above question, the author could provide the appropriate and full strain information of the {100} crystalline facets, determine whether it is tensile or compressive, and finally relate it to the catalytic performance of NPs, which is more reasonable.

Reviewer #2 (Remarks to the Author):

The authors have addressed almost all the points that I have raised. I have another two suggestions.

For the d-band theory, I think, the author should explain that whether this theory is related to the average lattice strain in three directions of the surface part of the grain? Or is it related to the average lattice strain along in-plane directions of the surface part of the grain?

Author reply:

Indeed, the change in the surface d-band center due to lattice distortion is induced by the tensile strain in three directions of the surface. The individual effect of the in-plane lattice strain was not investigated.

Changes in the manuscript:

Page 10, we modified the sentence:

This is explained by the d-band model, where the tensile strain leads to a narrowing of the d-band and an increased population of the latter, for late transition metals.

By:

This is explained by the d-band model, where the tensile strain (regardless of direction in the lattice) leads to a narrowing of the d-band and an increased population of the latter, for late transition metals.

“Only the map of the [111] component of the lattice strain tensor is presented. Indeed, to determine the full strain tensor, at least three or more non-parallel reflections must be collected. Multiple reflections from a single particle have been measured previously using BraggCDI, but such a study could not be carried out in this work due to the in situ experimental limitations of the operando cell. However, with the assumption of a cubic symmetry, it is sufficient to measure one Bragg reflection”

I agree to the viewpoint that “with the assumption of a cubic symmetry, it is sufficient to measure one Bragg reflection”. According to this viewpoint and the answer to the above question, the author could provide the appropriate and full strain information of the {100} crystalline facets, determine whether it is tensile or compressive, and finally relate it to the catalytic performance of NPs, which is more reasonable.

Author reply:

The remark of the reviewer 2 is well received and deserves clarifications. Indeed the sentence “ ... it is sufficient to measure one Bragg reflection” is misleading and modified as shown below. We presented the map of the magnitude of the [111] component of the lattice strain tensor, in other words we only have the distortion of the lattice in the 111 direction. From the latter we can compute the magnitude of du/dr_{111} but one can not infer the magnitude du/dr_{100} from the measurements we performed and presented in the manuscript. As stated in the manuscript, to determine the full strain tensor, at least three or more non-parallel reflections must be collected.

However, the 111 component is also sensitive to 100 lattice distortion, and so used throughout as a signature of the phenomenon occurring at the surface of the nanocrystal during the catalytic process, to image the elastic response.

Changes in the manuscript:

Page 7, we modified the sentence: “However, with the assumption of a cubic symmetry, it is sufficient to measure one Bragg reflection”

By:

However, the 111 component is also sensitive to 100, 110 lattice distortions, and so used throughout as a signature of the phenomenon occurring at the surface of the nanocrystals during the catalytic process, to image their elastic response.